# The Impact of Moderate-Intensity Continuous or High-Intensity Interval Training on Adipogenesis and Browning of Subcutaneous Adipose Tissue in Obese Male Rats

**DOI:** 10.3390/nu12040925

**Published:** 2020-03-27

**Authors:** Mousa Khalafi, Hamid Mohebbi, Michael E. Symonds, Pouran Karimi, Amir Akbari, Elma Tabari, Mehrsa Faridnia, Kamilia Moghaddami

**Affiliations:** 1Department of Exercise Physiology, Faculty of Physical Education and Sport Sciences, University of Guilan, Rasht 4199613776, Iran; mohebbi_h@yahoo.com (H.M.); amir.akbari58@yahoo.com (A.A.); tabarielma@gmail.com (E.T.); 2The Early Life Research Unit, Division of Child Health, Obstetrics and Gynaecology, and Nottingham Digestive Disease Centre and Biomedical Research Centre, School of Medicine, University of Nottingham, Nottingham NG7 2UH, UK; 3Neurosciences Research Center (NSRC), Tabriz University of Medical Sciences, Tabriz 5166614756, Iran; pouran.karimi@yahoo.com; 4Department of Exercise Physiology, University of Campus, University of Guilan, Rasht 4199613776, Iran; faridnia_mehrsa@yahoo.com; 5Department of pure and basic science, Hashtgerd Branch, Islamic Azad University, Karaj 3361659913, Iran; kamilia1351@yahoo.com

**Keywords:** exercise training, adipogenesis, white adipose tissue browning, obesity

## Abstract

This study compares the effect of two types of exercise training, i.e., moderate-intensity continuous training (MICT) or high-intensity interval training (HIIT) on the browning of subcutaneous white adipose tissue (scWAT) in obese male rats. Effects on fat composition, metabolites, and molecular markers of differentiation and energy expenditure were examined. Forty male Wistar rats were assigned to lean (*n* = 8) or obese (*n* = 32) groups and fed either a standard chow or high-fat obesogenic diet for 10 weeks. Eight lean and obese rats were then blood and tissue sampled, and the remaining obese animals were randomly allocated into sedentary, MICT, or HIIT (running on a treadmill 5 days/week) groups that were maintained for 12 weeks. Obesity increased plasma glucose and insulin and decreased irisin and FGF-21. In scWAT, this was accompanied with raised protein abundance of markers of adipocyte differentiation, i.e., C/EBP-α, C/EBP-β, and PPAR-γ, whereas brown fat-related genes, i.e., PRDM-16, AMPK/SIRT1/PGC-1α, were reduced as was UCP1 and markers of fatty acid transport, i.e., CD36 and CPT1. Exercise training increased protein expression of brown fat-related markers, i.e., PRDM-16, AMPK/SIRT1/PGC-1α, and UCP1, together with gene expression of fatty acid transport, i.e., CD36 and CPT1, but decreased markers of adipocyte differentiation, i.e., C/EBP-α, C/EBP-β, and plasma glucose. The majority of these adaptations were greater with HIIT compared to MICT. Our findings indicate that prolonged exercise training promotes the browning of white adipocytes, possibly through suppression of adipogenesis together with white to beige trans-differentiation and is dependent on the intensity of exercise.

## 1. Introduction

Obesity is the result of an imbalance between energy intake and energy consumption, characterized by the accumulation of excess adipose tissue (AT) and adipocyte dysfunction [1]. AT is a metabolically dynamic organ that modulates energy balance through the regulation of lipid and glucose metabolism [2]. Moreover, brown AT (BAT) can function as a metabolically active tissue contributing to non-shivering thermogenesis [2,3]. White (WAT) and BAT have different morphologies and precursor cells [4,5] and are classified by their size, the number of fat droplets, mitochondrial content, and the presence of uncoupling protein 1 (UCP1) [6,7]. In addition, beige adipocytes have been identified in WAT that are characterized as possessing more mitochondria than WAT, with enhanced gene expression for proteins involved in lipolysis and thermogenesis, such as UCP1 [8], together with a higher rate of fatty acid transport and oxidation [9].

There are numerous regulatory components and molecular mechanisms that control adipogenesis, of which peroxisome proliferator-activated receptor gamma (PPAR-γ) and CCAAT/enhancer-binding protein beta isoforms (C/EBPs) are the main transcription factors [10,11]. Peroxisome proliferator-activated receptor γ coactivator 1α (PGC-1α) is the main regulator of mitochondrial biogenesis and a primary activator of PPAR-γ [12,13], which is directly regulated by AMP-activated protein kinase (AMPK) and the SIRT1 pathway [14,15,16] and is upregulated by calorie restriction and exercise training [17,18]. Differentiation of brown and beige adipocytes is also regulated by PR domain containing 16 (PRDM16) [19,20], and the browning of WAT is accompanied with an increase in the abundance of cluster of differentiation 36 (CD36) and carnitine palmitoyltransferase 1 (CPT1), which facilitate the transfer of fatty acids (FAs) into cells and mitochondria, respectively [21,22]. 

Exercise training stimulates the PGC-1α pathway, and the abundance of brown adipocytes [18,23], and may induce the browning of WAT by promoting the release of muscle-derived myokines [24,25], although this remains controversial [26]. For example, irisin, a myokine which originates from the proteolytic cleavage of fibronectin type III domain-containing protein 5 (*FNDC5*), has been suggested to stimulate the expression of UCP1 in WAT and inhibit adipocyte differentiation [24,27]. In addition, fibroblast growth factor-21 (FGF-21) may promote browning through activation of the AMPK-SIRT-1-PGC-1α pathway [28]. The extent to which high-intensity interval training (HIIT), which is characterized by periods of high-intensity exercise with short rest intervals, may impact on beige fat with obesity is not established, although it can decrease fat-mass [29]. HIIT could be more effective than volume-matched moderate-intensity continuous training (MICT) in promoting beige fat, as well as reducing lipogenesis [30], improving oxygen uptake [31] and glucose metabolism [32], plus enhanced AMPK/PGC1-α gene expression [33] and raised irisin [34]. We hypothesize that the intensity of exercise training may induce different adaptations within scWAT after obesity that are related to the modulation of adipogenesis and white to beige trans-differentiation. 

## 2. Materials and Methods

### 2.1. Ethics Statement

All animal experiments were conducted according to the National Institute of Health ethical guidelines for the care and use of laboratory animals (NIH; Publication No. 85–23, revised 1985) and were reviewed and verified by the Veterinary Ethics Committee of Guilan University of Medical Sciences (Approval ID: IR.GUMS.REC.1397.081).

### 2.2. Animals and Diets

Forty (seven-week-old) male adult Wistar rats, weighing 160–200 g, were purchased from the Pasteur Institute Animal Care Center (Karaj, Iran). They were transferred to the Faculty of Physical Education and Sports Sciences of the University of Guilan and housed four per cage under a standard 12 h light/dark cycle at a constant temperature (25 + 2 °C) with ad-libitum access to food and water. After one week of acclimation, animals were randomly allocated into lean (L) (*n* = 8) or obese (Ob) (*n* = 32) groups and fed ad libitum either a standard chow or high-fat diet (HFD) until the end of each experiment (Figure 1). The HFD was fed to promote excess fat deposition and contained 60% fat (34.9 g/%), 20% carbohydrate (26.3 g/%), and 20% protein (26.2 g/%) [35], which compared with the normal chow diet that contained 10% fat (4.3 g/%), 70% carbohydrate (67.3 g/%), and 20% protein (19.2 g/%). 

### 2.3. Experimental Protocol

After 10 weeks, all lean animals and 8 Ob rats were sacrificed to determine the effect of obesity on specific metabolites and hormones, together with molecular markers of differentiation and browning. They were anesthetized with ketamine (60 mg/kg) and xylazine (6 mg/kg) injection, and scWAT was rapidly isolated and then either snap frozen and stored at −80 °C or fixed in 4% paraformaldehyde for histological examination. Blood samples were collected into ice-chilled ethylene diaminetetraacetic acid-containing tubes. Plasma was obtained by centrifugation at ~1830 g-force for 15 min at 4 °C within 15 min of collection and stored at –80 °C until subsequent analysis.

The remaining obese animals (*n* = 24) were randomly divided into three subgroups, which either remained sedentary (S) or were subjected to MICT or HIIT for a further 12 weeks and then tissue sampled.

### 2.4. Exercise Training

Maximum running speed was estimated using a treadmill (with a 25° inclination) [32,36]. The rats were placed on the treadmill and allowed to adapt over a 5-min period at a speed of 6 m/min. The speed was then gradually increased by 2 m/min every 2 min until they were unable or unwilling to continue. The running speed for Vo2max was defined as the maximal speed, for a set running distance. Exercise training was induced 5 days/week [37], of which the MICT protocol consisted of continuous running at a rate equivalent to 65%–70% of maximum speed, and the distance covered was matched to the HIIT protocol. Accordingly, the treadmill speed was gradually increased from 12 m/min in the first week, and reached 16 m/min in the 10th week and then maintained for the final two weeks. The HIIT protocol included 10 bouts of 4 min of high intensity running that reached 85%–90% of the maximal speed with 2 min active rest periods of 50% of maximal speed. The interval pace was increased gradually over the 10 weeks and maintained for the last two weeks. Accordingly, the treadmill speed was increased from 17 m/min in the first week and reached 26 m/min by the 10th week. Both HIIT and MICT protocols included 5 min of low-intensity (45%–50% of maximal speed) together with “warm-up” and “cool-down” periods before and after each session. 

### 2.5. Western Blotting

Protein lysates from scWAT were isolated using lysis buffer (50 mM Tris, pH 7.5, 150 mM sodium chloride, 1% NP-40, 0.5% sodium deoxycholate, 0.1% SDS, 0.1 mM EDTA and 0.1 mM EGTA) supplemented with complete protease inhibitor cocktail (Roche, Mannheim, Germany) and centrifuged at 12,000× *g* for 15 min at 4 °C. The protein concentration of the supernatant was determined by the Bradford method [38]. Proteins were separated using SDS–polyacrylamide gel electrophoresis using 8%–12% denatured ready gel (Bio-Rad, Hercules, CA, USA) and transferred onto a polyvinylidene difluoride (PVDF) membrane (Roche, West Sussex, UK). The membrane was blocked for 1 h in 5% BSA in tris-buffered saline and 0.1% Tween 20 (TBST) to block nonspecific bindings. Subsequently, blots were incubated overnight at 4 °C with primary antibodies: AMPKα1/2 (D-6) (sc-74461), p-AMPKα1-2 (sc-33524), C/EBPα(D-5) (sc-365318), C/EBP β (47A1) (sc-56637), PPARγ (E-8) (sc-7273), PRDM16 (N-16) (sc-130243), PGC1α (SC5815), SIRT1 (sc-74504), UCP1 (4E5) (sc-293418), ACCα(D-5) (sc-137104), and GAPDH (6C5) (sc-32233) (purchased from Santa Cruz (1:500)) and Acetyl-p53 (Lys382; purchased from cell Signaling Technology (1:500)). The membrane was then washed three times and incubated with the appropriate secondary antibody for 1 h at room temperature in 5% milk in TBST [38]. Protein bands were visualized with an enhanced chemiluminescence (ECL) reagent and radiographic film (Fuji) quantified by densitometry analysis with Image J software. With respect to the measurement of PGC1a protein, we pooled each of the bands from three gels into one, as suggested by the thermofisher protocol (http://tools.thermofisher.com/content/sfs/brochures/TR0051-Elute-from-polyacrylamide.pdf). Briefly, after separately running the electrophoresis on three separate gels and staining with coomassie Brilliant Blue (CBB), each lane the contained the appropriate molecular weight protein, i.e., bands just above and below 70–150 kda. The excised pieces of gel were then placed in microcentrifuge tubes and covered with 200 mL of elution buffer (50 mM Tris-HCl, 150 mM NaCl, and 0.1 mM EDTA; pH 7.5) and incubated in a rotary shaker at 30 °C overnight. These were then centrifuged at 12,000× *g* for 10 min and pipetted supernatant into a new microcentrifuge tube, after which the supernatant subjected to final SDS PAGE electrophoresis and Western blotting.

### 2.6. Histological Analysis and Immunofluorescence Staining

The paraffin-embedded scWAT samples were cut (5 µm) using a microtome and mounted on glass slides. These were then stained with hematoxylin-eosin (H&E) and light microscopy images taken over an area of at least 200 cells per section using Image J software v1.42q (National Institutes of Health) [39]. For immunostaining, slides were permeabilized in 0.1% Triton X in sodium phosphate buffer (PBS) and blocked in bovine plasma albumin (BSA; 3%) in PBS. They were incubated overnight with UCP1 antibody (sc-293418; Santa Cruz Biotechnology, Santa Cruz, CA, USA), diluted in 1% BSA solution, washed and incubated with secondary antibody for 45 min at 37 °C [40]. Images from at least three different areas were taken, and nuclei were counted by Image J analysis software.

### 2.7. Real-Time PCR Analysis

The ratio of mitochondrial DNA (mtDNA) to nuclear DNA (nDNA) was measured to determine the relative number of mitochondria in scWAT. For this purpose, total cellular DNA was extracted using the DNeasy blood and tissue kit and the relative levels of mtDNA and nDNA quantified using primers specific for mitochondrial 16sRNA (forward, 5′-CAGCTCGTG TCGTGAGATGT- 3′; reverse, 5′-CGTAAGGGCCATGATGACTT-3′) and the nuclear gene GAPDH (forward 5′-GCTGAACGGGAAACTCACTG-3′; reverse 5′-CCAGCATCGAAGGTAGAGGA-3′).

Total RNA content was extracted using Trizol (Invitrogen, Carlsbad, CA, USA) regent, then CDNA was synthesized using the iScript cDNA synthesis kit (Bio-Rad, Hercules, CA, USA). The relative gene expression of CD36 and CPT1 were evaluated by real-time quantitative polymerase chain reaction (PCR) amplification of the cDNAs using qPCR Mastermix plus for SYBR green using following primer sequences: CD36: 5′-ATGGGCTGTGATCGGAACTG-3′ and 5′-GTCTTCCCAATAAGCATGTCTCC- 3′; CPT1: 5′-CTCCTGGAAGAAACGCCTTATT-3′ and 5′-CACCTTGCAGTAGTTGGAACC-3′. Real-time PCR amplifications were performed with a Roche light cycler real-time PCR machine (Roche, Germany) [41]. The results were normalized to the level of glyceraldehyde 3-phosphate dehydrogenase (GAPDH) gene expression using the _ΔΔ_Ct method. 

### 2.8. Enzyme-Linked Immunosorbent Assay (ELISA)

ELISA kits (MyBioSources, Inc., San Diego, CA, USA) were used to measure plasma irisin (MBS2601445), FGF-21 (MBS030711), and insulin (MBS724709) according to the manufacturer’s instructions. Blood glucose concentration was determined immediately after sampling using a portable glucometer (FreeStyle OptiumH, Abbott Laboratories, Doncaster, Victoria, Australia).

### 2.9. Statistical Analysis

Data were analyzed using Graph Pad Prism 6.0 statistic software. A two-tailed *t*-test was used for comparison between normal and obese groups, and one-way ANOVA followed by Tukey post-hoc tests was performed for comparison of the effect of exercise training. All data were represented as the mean ± SEM. *p*-values < 0.05 were considered statistically significant. 

## 3. Results

### 3.1. Body Weight, Metabolites, Hormones and Exercise Performance

Obesity increased body weight, which was unaffected by exercise training and decreased plasma irisin and FGF-21 (Table 1). Both HIIT and MICT resulted in an increase in irisin and FGF-21 compared with obese animals. Moreover, obesity increased plasma insulin with exercise training decreasing glucose but not insulin. The maximal speed, running distance and time, was greater for HIIT compared with MICT (Table 2).

### 3.2. Adipocyte Size and Number

Adipocyte size increased, whereas the adipocyte number decreased with obesity (Figure 2), which decreased with exercise training. Obesity also decreased the abundance of unilocular adipocytes, whereas exercise training enhanced the number multilocular adipocytes.

### 3.3. Adipocyte Differentiation Biomarkers

In order to evaluate the effect of obesity and exercise training on adipocyte differentiation, protein abundance of biomarkers involved in the adipogenesis was measured, including C/EBP-α, C/EBP-β, PPAR-γ, and a master transcription factor regulating brown adipogenesis PRDM-16. Obesity increased the abundance of C/EBP-α, C/EBP-β, and PPAR-γ, whereas PRDM16 was decreased (Figure 3). In contrast, exercise training reduced the abundance of C/EBP-α, C/EBP-β, and increased PPAR-γ and PRDM-16, adaptations that were enhanced with HIIT.

### 3.4. UCP1 and Intracellular Energy Sensing

Obesity downregulated the abundance of SIRT1 and PGC-1α proteins and upregulated the abundance of SIRT1-p53 acetylation and ACC proteins but had no effect on p-AMPK (Figure 4), whereas exercise training increased the abundance of SIRT1, PGC-1α, and also p-AMPK and decreased the abundance of SIRT1-p53 acetylation and ACC proteins. The latter effect was enhanced with HIIT with respect to SIRT1, PGC-1α, and ACC (Figure 4). Obesity also decreased the abundance of UCP1, whereas exercise training resulted in an increase (Figure 5). 

### 3.5. Fatty Acid Transporters and mtDNA Content

Obesity increased gene expression of CD36 but did not affect CPT1 (Figure 6), whereas these were both, with exercise training, an effect that was greater with HIIT compared with MICT. Obesity had no effect on mtDNA, whereas exercise training resulted in an increase, an adaptation that was greater for HIIT compared with MICT. 

## 4. Discussion

We demonstrate that a majority of the adverse effects of obesity on scWAT can be reversed with prolonged exercise that was modulated by the intensity of training. This adaptation occurred in the absence of any weight loss, and although glucose was normalized, insulin was only reduced with HIIT. Obesity thus caused a diabetic-like state that, in scWAT, was characterized with a greater abundance of biomarkers for adipocyte differentiation (i.e., C/EBP-α, C/EBP-β, and PPAR-γ), fatty acid transport and oxidation together with a downregulation of markers of energy sensing (i.e., AMPK/SIRT1/PGC-1α) and UCP1, that were reversed with exercise. The extent of these positive adaptations within scWAT and the improved glucose homeostasis remains to be established as other fat depots are expected to show comparable responses [42]. Exercise training results in a range of endocrine adaptations and includes an increased release of myokines [43]. The extent to which individual myokines can impact on adipose tissue function remains controversial, especially with regard to irisin [44]. Acute exercise increases irisin [45] and FGF-21 [46], although this is dependent on the intensity of exercise [45], whereas obesity has the opposite effect [47]. We confirmed that plasma irisin and FGF-21 increased with exercise. FGF-21 can promote browning [25,48] through enhanced PPAR-γ activity [49] and can have an autocrine/paracrine effect on PGC-1α in AT [48]. Moreover, FGF-21 administration can lead to the deacetylation of PGC-1α through activation of SIRT1 and AMPK [28], although this response may be mediated through other pathways as it persists in mice lacking inducible adipocyte AMPK β1β2 [50]. The origin of FGF-21 is likely to be the liver [51] and can also result in an enhanced glucagon-insulin ratio [51], although skeletal muscle can also secrete FGF-21 [52]. 

Reduced adipocyte size and number with exercise training are expected to accompany weight loss [53] and concomitant increase in energy expenditure [54,55,56]. We found little change in body weight with exercise, although the histological changes within scWAT are indicative of a catabolic state. The loss of lipid from fat cells was accompanied by an upregulation of mRNA for CD36, an adaptation greater for HIIT than MICT, which is in accord with findings from obese patients [57]. Raised CD36, despite no change CPT1 that regulates cellular and mitochondrial FA transport [58], may promote FA delivery and oxidation [59,60].

Exercise training increased p-AMPK and SIRT1 in scWAT, with HIIT being more effective and in accord with the effect of swimming on the regulation of SIRT1, PGC-1α, and AMPK in skeletal muscle [24,33,61]. Complex networks of hormones and signaling pathways are involved in the regulation of energy expenditure [14,28] with activation of the AMPK/SIRT1/PGC-1α axis increasing energy expenditure and mitochondrial content [62]. AMPK and SIRT1 are considered as two important upstream regulators of PGC-1α and PPAR-γ and are linked to mitochondrial biogenesis, thermogenic gene expression, and adipocyte differentiation [14,28]. Indeed, AMPK [16,63,64,65] and SIRT1 [15,16] have similar regulatory roles in the activation of PGC-1α, and their activity can be increased during the mobilization of cellular energy stores with exercise training [66]. AMPK phosphorylation and the deacetylation of SIRT1 activate PGC-1α [14], can contribute to the browning of WAT, and increase thermogenesis through enhanced UCP1 [45,67,68]. Moreover, PGC-1α also triggers mitochondrial biogenesis, as seen with swim training [23]. The most likely explanation for enhanced beige adipocytes in scWAT is trans-differentiation of WAT, as illustrated by the marked downregulation of C/EBPα and C/EBPβ and upregulation of PRDM16 [69,70] and PPAR-γ [71], thereby overcoming the negative effects of obesity [72]. Future studies should consider in more detail the impact of exercise training on other fat depots together with proteomic response in order to gain a better overview of the range of pathways involved [73].

## 5. Conclusions

In conclusion, we demonstrate that the adverse effect of obesity on scWAT can be overcome with exercise training, with the magnitude of response determined by its intensity and possibly through suppression of adipogenesis together with white to beige trans-differentiation. 

## Figures and Tables

**Figure 1 nutrients-12-00925-f001:**
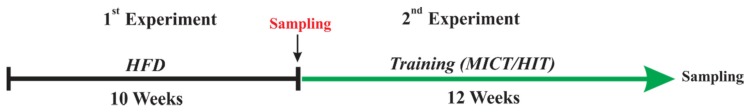
Experimental design. The first part of the experiment was designed for determining the effect of obesity (10 weeks) and the second part to determining the effect of 12 weeks of either moderate-intensity continuous training (MICT) or high-intensity interval training. (HIIT). HFD: high-fat diet.

**Figure 2 nutrients-12-00925-f002:**
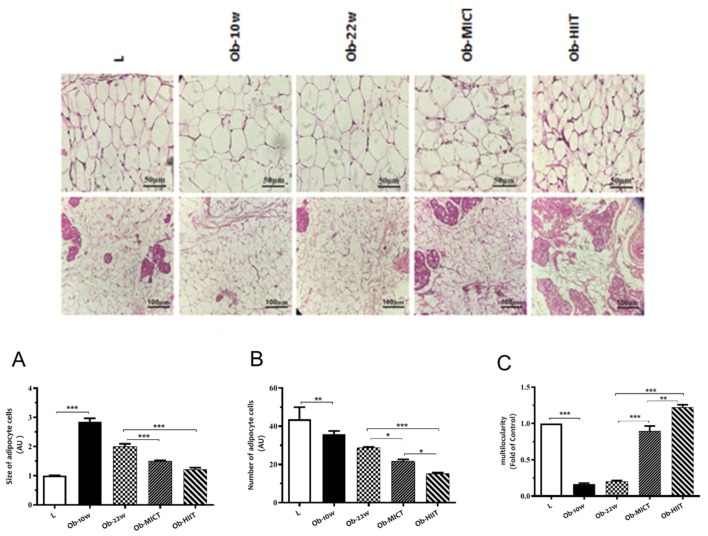
The effect of obesity and exercise training on adipocyte (**A**) size, (**B**) cell number, and (**C**) percentage of mulilocular cells in subcutaneous adipose tissue. Data are represented as means ± SEM (*n* = 6) and significant differences between groups indicated by: * *p* < 0.05, ** *p* < 0.01, *** *p* < 0.001. L: lean; Ob-10w: obesity-10 weeks; Ob-22w: obesity + sedentary; Ob-MICT: obesity + moderate-intensity continuous training; Ob-HIIT: obesity + high-intensity interval training (HIIT).

**Figure 3 nutrients-12-00925-f003:**
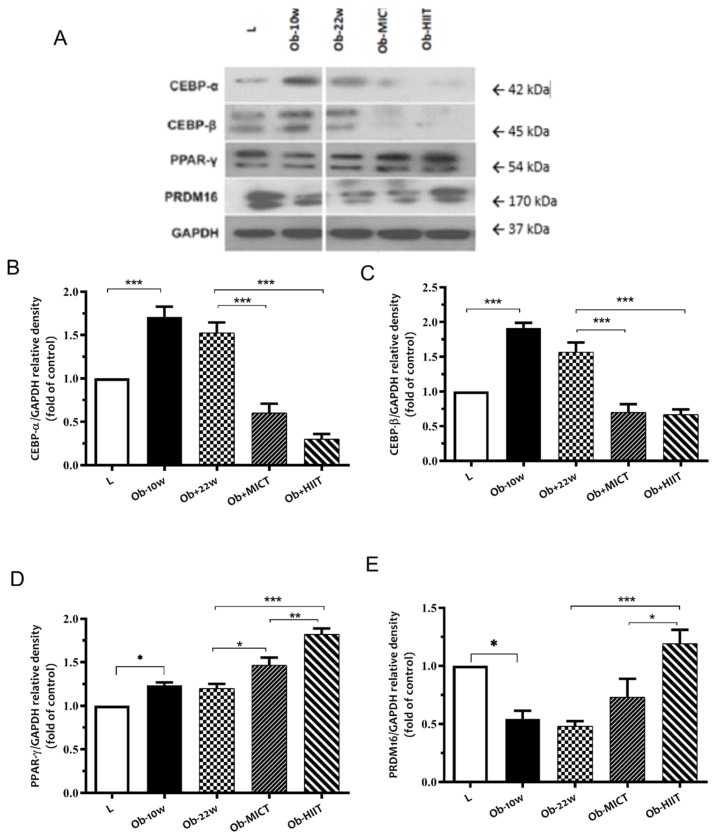
The effect of obesity and exercise training on markers of white adipocyte differentiation in subcutaneous adipose tissue. (**A**) Western blot analysis of protein expression for C/EBP-α, C/EBP-β, PPAR-γ, PRDM16, and GAPDH. Quantification graphs of (**B**) C/EBP-α, (**C**) C/EBP-β, (**D**) PPAR-γ, and (**E**) PRDM16 relative to GAPDH. Data are represented as means ± SEM (*n* = 6) and significant difference between groups indicated by * *p* < 0.05, ** *p* < 0.01, *** *p* < 0.001. L: lean; Ob-10w: obesity-10 weeks; Ob-22w: obesity + sedentary; Ob-MICT: obesity + moderate-intensity continuous training; Ob-HIIT: obesity + high-intensity interval training (HIIT).

**Figure 4 nutrients-12-00925-f004:**
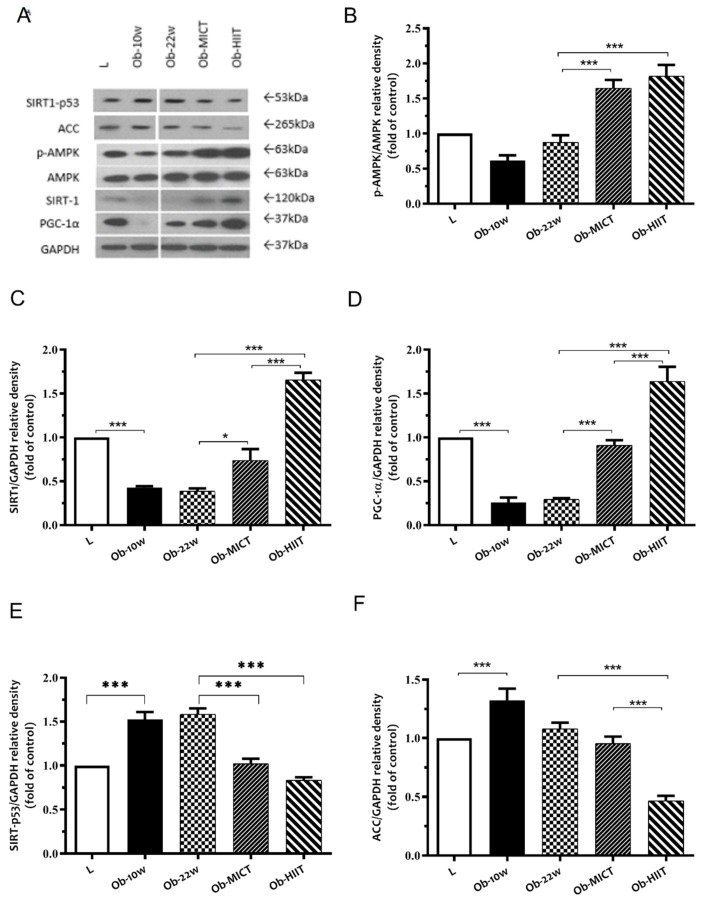
The effect of obesity and exercise training on protein expression of adipocyte differentiation markers in subcutaneous adipose tissue. (**A**) Western blot analysis of protein expression of p-AMPK, AMPK, SIRT1, PGC-1α, and GAPDH. Quantification graphs of (**B**) p-AMPK, (**C**) SIRT1, (**D**) PGC-1α, (**E**) SIRT1-p53, and (**F**) ACC proteins relative to GAPDH. Data are represented as means ± SEM (*n* = 6) and significant difference between groups indicated by * *p* < 0.05, *** *p* < 0.001. L: lean; Ob-10w: obesity-10 weeks; Ob-22w: obesity + sedentary; Ob-MICT: obesity + moderate-intensity continuous training; Ob-HIIT: obesity + high-intensity interval training (HIIT).

**Figure 5 nutrients-12-00925-f005:**
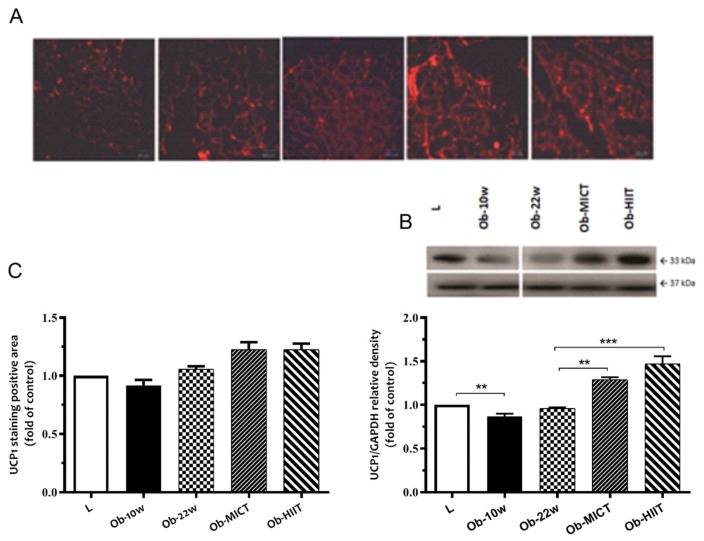
The effect of obesity and exercise training on uncoupling protein (UCP)1. (**A**) Immunofluorescence images of UCP1 positive cells in subcutaneous adipose tissue, (**B**) Western blot analysis for UCP1 and GAPDH, and (**C**) mean UCP1 abundance as determined by immunohistochemistry and Western blotting. Data are represented as means ± SEM (*n* = 6) and significant difference between groups indicated by ** *p* < 0.01, *** *p* < 0.001. L: lean; Ob-10w: obesity-10 weeks; Ob-22w: obesity + sedentary; Ob-MICT: obesity + moderate-intensity continuous training; Ob-HIIT: obesity + high-intensity interval training (HIIT).

**Figure 6 nutrients-12-00925-f006:**
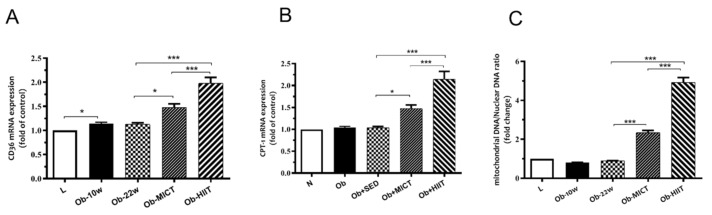
The effect of obesity and exercise training on mRNA expression of (**A**) CD36, (**B**) carnitine palmitoyltransferase I (CPT1), and (**C**) mitochondrial DNA content (mtDNA) in subcutaneous adipose tissue. Data are represented as means ± SEM and significant difference between groups indicated by * *p* < 0.05, *** *p* < 0.001. L: lean; Ob-10w: obesity-10 weeks; Ob-22w: obesity + sedentary; Ob-MICT: obesity + moderate-intensity continuous training; Ob-HIIT: obesity + high-intensity interval training (HIIT).

**Table 1 nutrients-12-00925-t001:** Effect of obesity and exercise training on body weight, plasma irisin, fibroblast growth factor (FGF)-21, glucose, and insulin.

Groups	Body Weight (gr)	Irisin (nmol/L)	FGF-21 (ng/L)	Insulin (ng/mL)	Glucose (mg/dL)	IR
L	274.00 ± 12.8	5.66 ± 0.59	1471.71 ± 124.54	1.65 ± 0.25	117.37 ± 1.93	2.30 ± 0.32
Ob-10w	366.62 ± 24.02 ***	4.41 ± 0.69 *	1080.00 ± 124.80 **	2.37 ± 0.36 ***	146.16 ± 24.30 **	4.14 ± 1.15 ***
Ob-22w	438.87 ± 21.88	4.61 ± 0.61	1093.83 ± 125.97	2.14 ± 0.43	154.66 ± 13.21	3.89 ± 0.54
Ob-MICT	407.87 ± 21.97	5.61 ± 0.39 *	1261.30 ± 45.57 **	1.82 ± 0.33	117.50 ± 1.37 ***	2.53 ± 0.46 **
Ob-HIIT	407.00 ± 42.98	6.43 ± 0.93 *	1658.83 ± 206.50 **	1.49 ± 0.15 *	111.16 ± 4.07 ***	1.96 ± 0.20 ***

Data are represented as means ± SEM and significant difference between groups indicated by * *p* < 0.05, ** *p* < 0.01,*** *p* < 0.001. L: lean; Ob-10w: obesity-10 weeks; Ob-22w: obesity + sedentary; Ob-MICT: obesity + moderate-intensity continuous training; Ob-HIIT: obesity + high intensity interval training (HIIT); IR: insulin resistance.

**Table 2 nutrients-12-00925-t002:** Effect of exercise training on exercise performance.

Groups	Maximal Speed (m.min^-1^)		Running Distance (m)		Running Time (min)	
	Pre	Post	Pre	Post	Pre	Post
Ob-22w	19.25 ± 1.83	19.00 ± 1.06	200.50 ± 31.26	196.75 ± 26.05	13.95 ± 1.52	13.77 ± 1.15
Ob-MICT	18.75 ± 1.83	27.50 ± 1.41 ***	197.37 ± 31.01	370.75 ± 30.87 ***	13.76 ± 1.50	22.46 ± 1.09 ***
Ob-HIIT	19.75 ± 1.66	30.25 ± 1.66 ***^##^	217.00 ± 35.34	434.62 ± 35.07 ***^##^	14.76 ± 1.62	24.7 ± 1.13 ***^##^

Data are represented as means ± SEM and significant difference between groups indicated by *** *p* < 0.01 and ^##^
*p* < 0.01. Ob-22w: obesity + sedentary; Ob-MICT: obesity + moderate-intensity continuous training; Ob-HIIT: obesity + high intensity interval training (HIIT).

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
