# Peer review of "The Impact of Moderate-Intensity Continuous or High-Intensity Interval Training on Adipogenesis and Browning of Subcutaneous Adipose Tissue in Obese Male Rats"

_nutrients, 2020, doi:10.3390/nu12040925_

Round 1

Reviewer 1 Report

Khalafi et al., investigated the effect of moderate-intensity continuous or high intensity interval training on adipogenesis and white 4 adipose tissue browning in rats.

The experiment contains a large amount of work and the experimental design is appropriate and interesting. Some of the presented findings are to some extent novel or at least interesting and the work contributes to the growing body of evidence suggesting that exercise training in rodents lead to indications of browning of white adipose tissue. The manuscript would greatly improve by addition of more physiological measurements to support the indications observed on protein level in the adipose tissue. A more thorough discussion of the data in relation to the existing literature and a more critical position to the limitation and validity of the analysis and measurements performed in the manuscript would be appropriate.

Below some comments to improve the manuscript.

Major comments

More physiological data supporting the different statements raised from measurements of proteins and genes would strengthen the impact of the findings and improve the manuscript massively. Measurements of oxygen consumption rate/energy expenditure, food intake, glucose uptake into SCAT, FA transport into SCAT, and plasma free fatty acid levels etc. One thing is that FGF21 is increased in plasma and AMPK activated in the adipose tissue, but if it is not associated with any physiological changes, the importance of these findings are not great. The statement of browning of the white adipose tissue does that lead to increased energy expenditure fx?

Body weight data as a minimum - and preferable also body composition data - after the interventions would help interpret the results. At least body weight is important to include in the manuscript.

In the discussion, the data could be discussed to a more thorough extent and related more to the existing literature. Please discuss what is found previously and how the observed findings are consistent with previous findings – but also what is in contrast to previous findings.

PGC-1a protein is very hard to measure in adipose tissue and skeletal muscle, since it is not very abundant. When measured with commercial or even self-developed antibodies, various unspecific bands are present. Even when measured on isolated nuclear fractions and loading a massive amount of protein, the specific band is extremely hard to detect. Please verify that your very strong and nice band with your commercial antibody is specific and a valid measure of PGC1a protein either by showing that this band disappears in PGC1a KO tissue or is overexpressed in cells/tissue overexpressing PGC1a. If not doable, I would suggest to take out those data.

After the abstract, I stopped correcting the language. I would suggest to have an English-speaking person to read the manuscript and thoroughly optimize the language in order to optimize the readability of the manuscript.

A measure of increased SIRT1 activity (fx. P53 acetylation) and potentially also SIRT1 substrate (NAD+) would strengthen the conclusion that exercise lead to increased SIRT1 signaling.

Plasma irisin levels are measured, but the measurement of irisin is highly debated. See fx PMID: 25749243 and https://www.sciencedirect.com/science/article/pii/S2212877820300247.  Please include a discussion of the measurement and emphasize arguments why your measurements are reliable and valid.

Minor comments:

Line 2-4: please emphasize that this has been done in rats.

Line 32 in abstract : CD36 and CPT1 – are they markers of energy expenditure?

Line 37: in → is

Line 54: PGC-1a has not previously been abbreviated.

“CPT1” and “CPT-1” are used inconsistently. Please be consistent with this abbreviation.

It is stated that FGF21 is activating AMPK and SIRT1. This is a bit controversial. See PMID:28580278. Consider discuss this matter.

AMPK protein – is that AMPKa1, AMPKa2, AMPKb1, AMPKb2, AMPKb, AMPKy1, AMPKy2, AMPKy3?

Moreover, since both phosphorylation level and protein content of AMPK seem to be different between conditions, consider whether showing phos and protein levels separately would be more appropriate and help the reader to interpret the data.

All over the manuscript, please differentiate between “exercise” as acute, one single session, and “exercise training” representing several continuous session of exercise bouts for a period.

Author Response

Referee 1

Major comments

More physiological data supporting the different statements raised from measurements of proteins and genes would strengthen the impact of the findings and improve the manuscript massively. Measurements of oxygen consumption rate/energy expenditure, food intake, glucose uptake into SCAT, FA transport into SCAT, and plasma free fatty acid levels etc. One thing is that FGF21 is increased in plasma and AMPK activated in the adipose tissue, but if it is not associated with any physiological changes, the importance of these findings are not great. The statement of browning of the white adipose tissue does that lead to increased energy expenditure fx?

Our response – the suggested additional measures would strengthen the paper but a majority of these were beyond the technical skills available for the study e.g. directly measuring metabolic flux across scWAT. We have added further results relating to the distance run by the animals – see Table 2. In addition, as suggested we have completed immunoblotting data of the acetylated p53 and ACCa to confirm the SIRT1 results – see Figure 4.

Body weight data as a minimum - and preferable also body composition data - after the interventions would help interpret the results. At least body weight is important to include in the manuscript.

Our response - body weight data has been added as requested – see Table 1.

 In the discussion, the data could be discussed to a more thorough extent and related more to the existing literature. Please discuss what is found previously and how the observed findings are consistent with previous findings – but also what is in contrast to previous findings.

Our response – the discussion has been rewritten.

PGC-1a protein is very hard to measure in adipose tissue and skeletal muscle, since it is not very abundant. When measured with commercial or even self-developed antibodies, various unspecific bands are present. Even when measured on isolated nuclear fractions and loading a massive amount of protein, the specific band is extremely hard to detect. Please verify that your very strong and nice band with your commercial antibody is specific and a valid measure of PGC1a protein either by showing that this band disappears in PGC1a KO tissue or is overexpressed in cells/tissue overexpressing PGC1a. If not doable, I would suggest to take out those data.

Our response - PGC1a KO tissue or overexpressed in cells/tissue overexpressing PGC1a is not currently unavailable to us, but we did undertake additional validation steps that we have added to the methods – see lines 145-153 - according to the thermofisher protocol (http://tools.thermofisher.com/content/sfs/brochures/TR0051-Elute-from-polyacrylamide.pdf).

After the abstract, I stopped correcting the language. I would suggest to have an English-speaking person to read the manuscript and thoroughly optimize the language in order to optimize the readability of the manuscript.

Our response – the manuscript has been rewritten throughout in order to correct the English.

A measure of increased SIRT1 activity (fx. P53 acetylation) and potentially also SIRT1 substrate (NAD+) would strengthen the conclusion that exercise lead to increased SIRT1 signaling.

Our response – we have added immunoblotting data of the acetylated p53 and ACCa to confirm the SIRT1 activity and lipogenesis as suggested – see Figure 4.

Plasma irisin levels are measured, but the measurement of irisin is highly debated.

See fx PMID: 25749243 and https://www.sciencedirect.com/science/article/pii/S2212877820300247.  Please include a discussion of the measurement and emphasize arguments why your measurements are reliable and valid.

Our response – we have rewritten this part of the discussion as suggested – see lines 277-282.

Minor comments:

Line 2-4: please emphasize that this has been done in rats.

Line 32 in abstract: CD36 and CPT1 – are they markers of energy expenditure?

Line 37: in → is

Line 54: PGC-1a has not previously been abbreviated.

“CPT1” and “CPT-1” are used inconsistently. Please be consistent with this abbreviation.

Our response – amended as requested.

It is stated that FGF21 is activating AMPK and SIRT1. This is a bit controversial. See PMID:28580278. Consider discuss this matter.

Our response: we have rewritten this section of the discussion as suggested.

AMPK protein – is that AMPKa1, AMPKa2, AMPKb1, AMPKb2, AMPKb, AMPKy1, AMPKy2, AMPKy3?

Our response: we used p-AMPKα1/2 antibody (Thr 172): sc-33524 & AMPKα1/2 antibody (D-6): sc-74461 by immunoblotting and detected a single band corresponding at 63 kDa – see Figure 4a.

Moreover, since both phosphorylation level and protein content of AMPK seem to be different between conditions, consider whether showing phos and protein levels separately would be more appropriate and help the reader to interpret the data.

Our response – the results presented are in Figure 4b are the ratio of pAMPK to AMPK protein. This is in accord with other comparable studies. For example, PMID: 22623023 by Łukaszuk et al 2012, Lipids.

All over the manuscript, please differentiate between “exercise” as acute, one single session, and “exercise training” representing several continuous session of exercise bouts for a period.

Our response – corrected as necessary although “exercise” can refer to several sessions.

Reviewer 2 Report

nutrients-714740

Title : The impact of moderate-intensity continuous or high-intensity interval training on adipogenesis and white adipose tissue browning

The authors present a well-performed work to evaluate how high intensity interval training (HIIT) compared to moderate intensity continuous exercise training (MICT) is able to impact adipogenesis and white adipose tissue metabolic status in rats. They reported that both HIIT and MICT are efficient to modulate adipocytes size and white to beige trans-differentiation. The study is well designed and the results support the conclusion. We can however regret that the study remain descriptive with no proof of concept. We can also regret the lack of novelty (see the work of Wang et al., Life of Sciences 2017).

Major comments:

  • No data are provided regarding body weight and composition (fat accumulation) of animals all along the study. Indeed, Wistar rats were not a very accurate model of obesity in rodent.
  • The authors stipulate that adipocyte number decrease with obesity. This is unexpected? It seems that the method used to evaluate the number of adipocytes is not accurate.
  • We can regret that only biochemical data are provided with no functional evaluation, such as insulin or glucose tolerance test, adipocytes metabolic activities, maximal aerobic velocities…
  • The novelty of this work, which not so obvious, clearly need to be emphasized.

Author Response

Referee 2

The authors present a well-performed work to evaluate how high intensity interval training (HIIT) compared to moderate intensity continuous exercise training (MICT) is able to impact adipogenesis and white adipose tissue metabolic status in rats. They reported that both HIIT and MICT are efficient to modulate adipocytes size and white to beige trans-differentiation. The study is well designed and the results support the conclusion. We can however regret that the study remain descriptive with no proof of concept. We can also regret the lack of novelty (see the work of Wang et al., Life of Sciences 2017).

Major comments:

  • No data are provided regarding body weight and composition (fat accumulation) of animals all along the study. Indeed, Wistar rats were not a very accurate model of obesity in rodent.

Our response: Body weight data has been added to Table 1 – showing a clear increase with consumption of the HFD. Individual fat depots were not dissected.

  • The authors stipulate that adipocyte number decrease with obesity. This is unexpected? It seems that the method used to evaluate the number of adipocytes is not accurate.

Our response, we measured the number and size of adipocytes over a fixed area hence the finding or larger, but fewer cells in consistent.

  • We can regret that only biochemical data are provided with no functional evaluation, such as insulin or glucose tolerance test, adipocytes metabolic activities, maximal aerobic velocities…

Our response, we have added exercise performance parameters including maximal speed, running distance and time in Table 2.

  • The novelty of this work, which not so obvious, clearly need to be emphasized.

Our response – we have rewritten the dicussion to emphasise the novelty of the study.

Round 2

Reviewer 2 Report

The manuscript has been significantly improved.

No additional comment